# Is Q-learning Provably Efficient?

**Chi Jin**[*]
University of California, Berkeley
chijin@cs.berkeley.edu

**Zeyuan Allen-Zhu**[*]
Microsoft Research, Redmond
zeyuan@csail.mit.edu

**Sebastien Bubeck**
Microsoft Research, Redmond
sebubeck@microsoft.com

**Michael I. Jordan**
University of California, Berkeley
jordan@cs.berkeley.edu

## Abstract

Model-free reinforcement learning (RL) algorithms, such as Q-learning, directly parameterize and update value functions or policies without explicitly modeling the environment. They are typically simpler, more flexible to use, and thus more prevalent in modern deep RL than model-based approaches. However, empirical work has suggested that model-free algorithms may require more samples to learn [7, 22]. The theoretical question of "whether model-free algorithms can be made *sample efficient*" is one of the most fundamental questions in RL, and remains unsolved even in the basic scenario with finitely many states and actions.

We prove that, in an episodic MDP setting, Q-learning with UCB exploration achieves regret $\tilde{\mathcal{O}}(\sqrt{H^3 S A T})$, where $S$ and $A$ are the numbers of states and actions, $H$ is the number of steps per episode, and $T$ is the total number of steps. This sample efficiency matches the optimal regret that can be achieved by any model-based approach, up to a single $\sqrt{H}$ factor. To the best of our knowledge, this is the first analysis in the model-free setting that establishes $\sqrt{T}$ regret *without* requiring access to a "simulator."

## 1 Introduction

Reinforcement Learning (RL) is a control-theoretic problem in which an agent tries to maximize its cumulative rewards via interacting with an unknown *environment* through time [26]. There are two main approaches to RL: model-based and model-free. Model-based algorithms make use of a model for the environment, forming a control policy based on this learned model. Model-free approaches dispense with the model and directly update the *value function*—the expected reward starting from each state, or the *policy*—the mapping from states to their subsequent actions. There has been a long debate on the relative pros and cons of the two approaches [7].

From the classical Q-learning algorithm [27] to modern DQN [17], A3C [18], TRPO [22], and others, most state-of-the-art RL has been in the model-free paradigm. Its pros—model-free algorithms are online, require less space, and, most importantly, are more expressive since specifying the value functions or policies is often more flexible than specifying the model for the environment—arguably outweigh its cons relative to model-based approaches. These relative advantages underly the significant successes of model-free algorithms in deep RL applications [17, 24].

On the other hand it is believed that model-free algorithms suffer from a higher sample complexity compared to model-based approaches. This has been evidenced empirically in [7, 22], and recent work has tried to improve the sample efficiency of model-free algorithms by combining them with

---

[*]The first two authors contributed equally. Full paper (and future edition) available at https://arxiv.org/abs/1807.03765.

model-based approaches [19, 21]. There is, however, little theory to support such blending, which requires a more quantitative understanding of relative sample complexities. Indeed, the following basic theoretical questions remain open:

**Can we design model-free algorithms that are sample efficient?**
In particular, **is Q-learning provably efficient?**

The answers remain elusive even in the basic tabular setting where the number of states and actions are finite. In this paper, we attack this problem head-on in the setting of the episodic Markov Decision Process (MDP) formalism (see Section 2 for a formal definition). In this setting, an episode consists of a run of MDP dynamics for $H$ steps, where the agent aims to maximize total reward over multiple episodes. We do not assume access to a "simulator" (which would allow us to query arbitrary state-action pairs of the MDP) and the agent is not allowed to "reset" within each episode. This makes our setting sufficiently challenging and realistic. In this setting, the standard Q-learning heuristic of incorporating $\varepsilon$-greedy exploration appears to take exponentially many episodes to learn [14].

As seen in the literature on bandits, the key to achieving good sample efficiency generally lies in managing the tradeoff between *exploration* and *exploitation*. One needs an efficient strategy to explore the uncertain environment while maximizing reward. In the model-based setting, a recent line of research has imported ideas from the bandit literature—including the use of upper confidence bounds (UCB) and improved design of exploration bonuses—and has obtained asymptotically optimal sample efficiency [1, 5, 10, 12]. In contrast, the understanding of model-free algorithms is still very limited. To the best of our knowledge, the only existing theoretical result on model-free RL that applies to the episodic setting is for *delayed Q-learning*; however, this algorithm is quite sample-inefficient compared to model-based approaches [25].

In this paper, we answer the two aforementioned questions affirmatively. We show that Q-learning, when equipped with a UCB exploration policy that incorporates estimates of the confidence of Q values and assign exploration bonuses, achieves total regret $\tilde{\mathcal{O}}(\sqrt{H^3 SAT})$. Here, $S$ and $A$ are the numbers of states and actions, $H$ is the number of steps per episode, and $T$ is the total number of steps. Up to a $\sqrt{H}$ factor, our regret matches the information-theoretic optimum, which can be achieved by model-based algorithms [5, 12]. Since our algorithm is just Q-learning, it is online and does not store additional data besides the table of Q values (and a few integers per entry of this table). Thus, it also enjoys a significant advantage over model-based algorithms in terms of time and space complexities. To our best knowledge, this is the first sharp analysis for model-free algorithms—featuring $\sqrt{T}$ regret or equivalently $O(1/\varepsilon^2)$ samples for $\varepsilon$-optimal policy—*without* requiring access to a "simulator."

For practitioners, there are two key takeaways from our theoretical analysis:

1. The use of UCB exploration instead of $\varepsilon$-greedy exploration in the model-free setting allows for better treatment of uncertainties for different states and actions.
2. It is essential to use a learning rate which is $\alpha_t = O(H/t)$, instead of $1/t$, when a state-action pair is being updated for the $t$-th time. The former learning rate assigns more weight to updates that are more recent, as opposed to assigning uniform weights to all previous updates. This delicate choice of reweighting leads to the crucial difference between our sample-efficient guarantee versus earlier highly inefficient results that require exponentially many samples in $H$.

## 1.1 Related Work

In this section, we focus our attention on theoretical results for the tabular MDP setting, where the numbers of states and actions are finite. We acknowledge that there has been much recent work in RL for continuous state spaces [see, e.g., 9, 11], but this setting is beyond our scope.

**With simulator.** Some results assume access to a simulator [15] (a.k.a., a generative model [3]), which is a strong oracle that allows the algorithm to query arbitrary state-action pairs and return the reward and the next state. The majority of these results focus on an infinite-horizon MDP with discounted reward [e.g., 2, 3, 8, 16, 23]. When a simulator is available, model-free algorithms [2] (variants of Q-learning) are known to be almost as sample efficient as the best model-based algorithms [3]. However, the simulator setting is considered to much easier than standard RL, as it "does not require exploration" [2]. Indeed, a naive exploration strategy which queries all state-action

| | Algorithm | Regret | Time | Space |
|---|---|---|---|---|
| Model-based | RLSVI [? ] | $\tilde{\mathcal{O}}(\sqrt{H^3SAT})$ | $\tilde{\mathcal{O}}(TS^2A^2)$ | $\mathcal{O}(S^2A^2H)$ |
| | UCRL2 [10] [1] | at least $\tilde{\mathcal{O}}(\sqrt{H^4S^2AT})$ | $\Omega(TS^2A)$ | $\mathcal{O}(S^2AH)$ |
| | Agrawal and Jia [1] [1] | at least $\tilde{\mathcal{O}}(\sqrt{H^3S^2AT})$ | | |
| | UCBVI [5] [2] | $\tilde{\mathcal{O}}(\sqrt{H^2SAT})$ | $\tilde{\mathcal{O}}(TS^2A)$ | |
| | vUCQ [12] [2] | $\tilde{\mathcal{O}}(\sqrt{H^2SAT})$ | | |
| Model-free | Q-learning ($\varepsilon$-greedy) [14] (if 0 initialized) | $\Omega(\min\{T, A^{H/2}\})$ | $\mathcal{O}(T)$ | $\mathcal{O}(SAH)$ |
| | Delayed Q-learning [25] [3] | $\tilde{\mathcal{O}}_{S,A,H}(T^{4/5})$ | | |
| | Q-learning (UCB-H) | $\tilde{\mathcal{O}}(\sqrt{H^4SAT})$ | | |
| | Q-learning (UCB-B) | $\tilde{\mathcal{O}}(\sqrt{H^3SAT})$ | | |
| | lower bound | $\Omega(\sqrt{H^2SAT})$ | - | - |

Table 1: Regret comparisons for RL algorithms on episodic MDP. $T = KH$ is totally number of steps, $H$ is the number of steps per episode, $S$ is the number of states, and $A$ is the number of actions. For clarity, this table is presented for $T \geq \text{poly}(S, A, H)$, omitting low order terms.

pairs uniformly at random already leads to the most efficient algorithm for finding optimal policy [3].

**Without simulator.** Reinforcement learning becomes much more challenging without the presence of a simulator, and the choice of exploration policy can now determine the behavior of the learning algorithm. For instance, Q-learning with $\varepsilon$-greedy may take exponentially many episodes to learn the optimal policy [14] (for the sake of completeness, we present this result in our episodic language in Appendix A).

Mathemtically, this paper defines "model-free" algorithms as in existing literature [25, 26]:

**Definition 1.** *A reinforcement learning algorithm is **model-free** if its space complexity is* always sublinear *(for any $T$) relative to the space required to store an MDP. In episodic setting of this paper, a model-free algorithm has space complexity $o(S^2AH)$ (independent of $T$).*

In the model-based setting, UCRL2 [10] and Agrawal and Jia [1] form estimates of the transition probabilities of the MDP using past samples, and add upper-confidence bounds (UCB) to the estimated transition matrix. When applying their results to the episodic MDP scenario, their total regret is at least $\tilde{\mathcal{O}}(\sqrt{H^4S^2AT})$ and $\tilde{\mathcal{O}}(\sqrt{H^3S^2AT})$ respectively.[1] In contrast, the information-theoretic lower bound is $\tilde{\mathcal{O}}(\sqrt{H^2SAT})$. The additional $\sqrt{S}$ and $\sqrt{H}$ factors were later removed by the UCBVI algorithm [5] which adds a UCB bonus directly to the Q values instead of the estimated transition matrix.[2] The vUCQ algorithm [12] is similar to UCBVI but improves lower-order regret terms using variance reduction. Finally, RLSVI [? ], an algorithm designed for setting of linear approxmation, provides $\tilde{\mathcal{O}}(\sqrt{H^3SAT})$ regret bound when adapted to tabular MDP setting. However, it is batch algorithm in nature, and requires $O(d^2H)$ space where in tabular setting $d = SA$.

We note that despite the sharp regret guarantees, most of the results in this line of research crucially rely on estimating and storing the entire transition matrix and thus suffer from unfavorable time and space complexities compared to model-free algorithms.

In the model-free setting, Strehl et al. [25] introduced delayed Q-learning, where, to find an $\varepsilon$-optimal policy, the Q value for each state-action pair is updated only once every $m = \tilde{\mathcal{O}}(1/\varepsilon^2)$ times this pair is visited. In contrast to the incremental update of Q-learning, delayed Q-learning always replaces old Q values with the average of the most recent $m$ experiences. When translated to the setting of this paper, this gives $\tilde{\mathcal{O}}(T^{4/5})$ total regret, ignoring factors in $S, A$ and $H$.[3] This is quite suboptimal compared to the $\tilde{\mathcal{O}}(\sqrt{T})$ regret achieved by model-based algorithm.

## 2 Preliminary

We consider the setting of a tabular episodic Markov decision process, $\mathrm{MDP}(\mathcal{S}, \mathcal{A}, \mathrm{H}, \mathbb{P}, \mathrm{r})$, where $\mathcal{S}$ is the set of states with $|\mathcal{S}| = S$, $\mathcal{A}$ is the set of actions with $|\mathcal{A}| = A$, $H$ is the number of steps in each episode, $\mathbb{P}$ is the transition matrix so that $\mathbb{P}_h(\cdot|x, a)$ gives the distribution over states if action $a$ is taken for state $x$ at step $h \in [H]$, and $r_h : \mathcal{S} \times \mathcal{A} \to [0, 1]$ is the deterministic reward function at step $h$.[4]

In each episode of this MDP, an initial state $x_1$ is picked arbitrarily by an adversary. Then, at each step $h \in [H]$, the agent observes state $x_h \in \mathcal{S}$, picks an action $a_h \in \mathcal{A}$, receives reward $r_h(x_h, a_h)$, and then transitions to a next state, $x_{h+1}$, that is drawn from the distribution $\mathbb{P}_h(\cdot|x_h, a_h)$. The episode ends when $x_{H+1}$ is reached.

A policy $\pi$ of an agent is a collection of $H$ functions $\left\{ \pi_h : \mathcal{S} \to \mathcal{A} \right\}_{h \in [H]}$. We use $V_h^\pi : \mathcal{S} \to \mathbb{R}$ to denote the value function at step $h$ under policy $\pi$, so that $V_h^\pi(x)$ gives the expected sum of remaining rewards received under policy $\pi$, starting from $x_h = x$, until the end of the episode. In symbols:

$$V_h^\pi(x) := \mathbb{E}\left[ \sum_{h'=h}^H r_{h'}(x_{h'}, \pi_{h'}(x_{h'})) | x_h = x \right] .$$

Accordingly, we also define $Q_h^\pi : \mathcal{S} \times \mathcal{A} \to \mathbb{R}$ to denote $Q$-value function at step $h$ so that $Q_h^\pi(x, a)$ gives the expected sum of remaining rewards received under policy $\pi$, starting from $x_h = x, a_h = a$, till the end of the episode. In symbols:

$$Q_h^\pi(x, a) := r_h(x, a) + \mathbb{E}[\sum_{h'=h+1}^H r_{h'}(x_{h'}, \pi_{h'}(x_{h'})) | x_h = x, a_h = a] .$$

Since the state and action spaces, and the horizon, are all finite, there always exists (see, e.g., [5]) an optimal policy $\pi^\star$ which gives the optimal value $V_h^\star(x) = \sup_\pi V_h^\pi(x)$ for all $x \in \mathcal{S}$ and $h \in [H]$. For simplicity, we denote $[\mathbb{P}_h V_{h+1}](x, a) := \mathbb{E}_{x' \sim \mathbb{P}(\cdot|x, a)} V_{h+1}(x')$. Recall the Bellman equation and the Bellman optimality equation:

$$\begin{cases} V_h^\pi(x) = Q_h^\pi(x, \pi_h(x)) \\ Q_h^\pi(x, a) := (r_h + \mathbb{P}_h V_{h+1}^\pi)(x, a) \\ V_{H+1}^\pi(x) = 0 \qquad \forall x \in \mathcal{S} \end{cases} \quad \text{and} \quad \begin{cases} V_h^\star(x) = \max_{a \in \mathcal{A}} Q_h^\star(x, a) \\ Q_h^\star(x, a) := (r_h + \mathbb{P}_h V_{h+1}^\star)(x, a) \qquad (2.1) \\ V_{H+1}^\star(x) = 0 \qquad \forall x \in \mathcal{S} . \end{cases}$$

The agent plays the game for $K$ episodes $k = 1, 2, \ldots, K$, and we let the adversary pick a starting state $x_1^k$ for each episode $k$, and let the agent choose a policy $\pi_k$ before starting the $k$-th episode. The total (expected) regret is then

$$\mathrm{Regret}(K) = \sum_{k=1}^K \left[ V_1^\star(x_1^k) - V_1^{\pi_k}(x_1^k) \right] .$$

## 3 Main Results

In this section, we present our main theoretical result—a sample complexity result for a variant of Q-learning that incorporates UCB exploration. We also present a theorem that establishes an information-theoretic lower bound for episodic MDP.

As seen in the bandit setting, the choice of exploration policy plays an essential role in the efficiency of a learning algorithm. In episodic MDP, Q-learning with the commonly used $\varepsilon$-greedy exploration strategy can be very inefficient: it can take exponentially many episodes to learn [14]

**Algorithm 1** Q-learning with UCB-Hoeffding

---
1: initialize $Q_h(x,a) \leftarrow H$ and $N_h(x,a) \leftarrow 0$ for all $(x,a,h) \in \mathcal{S} \times \mathcal{A} \times [H]$.
2: **for** episode $k = 1, \dots, K$ **do**
3:      receive $x_1$.
4:      **for** step $h = 1, \dots, H$ **do**
5:          Take action $a_h \leftarrow \operatorname{argmax}_{a'} Q_h(x_h, a')$, and observe $x_{h+1}$.
6:          $t = N_h(x_h, a_h) \leftarrow N_h(x_h, a_h) + 1$; $b_t \leftarrow c\sqrt{H^3\iota/t}$.
7:          $Q_h(x_h, a_h) \leftarrow (1 - \alpha_t)Q_h(x_h, a_h) + \alpha_t[r_h(x_h, a_h) + V_{h+1}(x_{h+1}) + b_t]$.
8:          $V_h(x_h) \leftarrow \min\{H, \max_{a' \in \mathcal{A}} Q_h(x_h, a')\}$.

---

(see also Appendix A). In contrast, our algorithm (Algorithm 1), which is Q-learning with an upper-confidence bound (UCB) exploration strategy, will be seen to be efficient. This algorithm maintains Q values, $Q_h(x,a)$, for all $(x,a,h) \in \mathcal{S} \times \mathcal{A} \times [H]$ and the corresponding V values $V_h(x) \leftarrow \min\{H, \max_{a' \in \mathcal{A}} Q_h(x, a')\}$. If, at time step $h \in [H]$, the state is $x \in \mathcal{S}$, the algorithm takes the action $a \in \mathcal{A}$ that maximizes the current estimate $Q_h(x,a)$, and is apprised of the next state $x' \in \mathcal{S}$. The algorithm then updates the Q values:

$$Q_h(x,a) \leftarrow (1 - \alpha_t)Q_h(x,a) + \alpha_t[r_h(x,a) + V_{h+1}(x') + b_t] \ ,$$

where $t$ is the counter for how many times the algorithm has visited the state-action pair $(x,a)$ at step $h$, $b_t$ is the confidence bonus indicating how certain the algorithm is about current state-action pair, and $\alpha_t$ is a learning rate defined as follows:

$$\alpha_t := \frac{H+1}{H+t} \ . \tag{3.1}$$

As mentioned in the introduction, our choice of learning rate $\alpha_t$ scales as $O(H/t)$ instead of $O(1/t)$—this is crucial to obtain regret that is not exponential in $H$.

We present analyses for two different specifications of the upper confidence bonus $b_t$ in this paper:

**Q-learning with Hoeffding-style bonus.** The first (and simpler) choice is $b_t = O(\sqrt{H^3\iota/t})$. (Here, and throughout this paper, we use $\iota := \log(SAT/p)$ to denote a log factor.) This choice of bonus makes sense intuitively because: (1) Q-values are upper-bounded by $H$, and, accordingly, (2) Hoeffding-type martingale concentration inequalities imply that if we have visited $(x,a)$ for $t$ times, then a confidence bound for the Q value scales as $1/\sqrt{t}$. For this reason, we call this choice *UCB-Hoeffding* (UCB-H). See Algorithm 1.

**Theorem 2** (Hoeffding). *There exists an absolute constant $c > 0$ such that, for any $p \in (0,1)$, if we choose $b_t = c\sqrt{H^3\iota/t}$, then with probability $1 - p$, the total regret of Q-learning with UCB-Hoeffding (see Algorithm 1) is at most $O(\sqrt{H^4SAT\iota})$, where $\iota := \log(SAT/p)$.*

Theorem 2 shows, under a rather simple choice of exploration bonus, Q-learning can be made very efficient, enjoying a $\tilde{O}(\sqrt{T})$ regret which is optimal in terms of dependence on $T$. To the best of our knowledge, this is the first analysis of a model-free procedure that features a $\sqrt{T}$ regret *without* requiring access to a "simulator."

Compared to the previous model-based results, Theorem 2 shows that the regret (or equivalently the sample complexity; see discussion in full version) of this version of Q-learning is as good as the best model-based one in terms of the dependency on the number of states $S$, actions $A$ and the total number of steps $T$. Although our regret slightly increases the dependency on $H$, the algorithm is online and does not store additional data besides the table of Q values (and a few integers per entry of this table). Thus, it enjoys an advantage over model-based algorithms in time and space complexities, especially when the number of states $S$ is large.

**Q-learning with Bernstein-style bonus.** Our second specification of $b_t$ makes use of a Bernstein-style upper confidence bound. The key observation is that, although in the worst case the value function is at most $H$ for any state-action pair, if we sum up the "total variance of the value function" for an entire episode, we obtain a factor of only $O(H^2)$ as opposed to the naive $O(H^3)$ bound (see Lemma C.5). This implies that the use of a Bernstein-type martingale concentration result could be

sharper than the Hoeffding-type bound by an additional factor of $H$.[5] (The idea of using Bernstein instead of Hoeffding for reinforcement learning applications has appeared in previous work; see, e.g., [3, 4, 16].)

Using Bernstein concentration requires us to design the bonus term $b_t$ more carefully, as it now depends on the empirical variance of $V_{h+1}(x')$ where $x'$ is the next state over the previous $t$ visits of current state-action $(x, a)$. This empirical variance can be computed in an online fashion without increasing the space complexity of Q-learning. We defer the full specification of $b_t$ to Algorithm 2 in Appendix C. We now state the regret theorem for this approach.

**Theorem 3** (Bernstein). *For any $p \in (0, 1)$, one can specify $b_t$ so that with probability $1-p$, the total regret of Q-learning with UCB-Bernstein (see Algorithm 2) is at most $O(\sqrt{H^3 SAT\iota} + \sqrt{H^9 S^3 A^3} \cdot \iota^2)$.*

Theorem 3 shows that for Q-learning with UCB-B exploration, the leading term in regret (which scales as $\sqrt{T}$) improves by a factor of $\sqrt{H}$ over UCB-H exploration, at the price of using a more complicated exploration bonus design. The asymptotic regret of UCB-B is now only one $\sqrt{H}$ factor worse than the best regret achieved by model-based algorithms.

We also note that Theorem 3 has an additive term $O(\sqrt{H^9 S^3 A^3} \cdot \iota^2)$ in its regret, which dominates the total regret when $T$ is not very large compared with $S, A$ and $H$. It is not clear whether this lower-order term is essential, or is due to technical aspects of the current analysis.

**Information-theoretical limit.** To demonstrate the sharpness of our results, we also note an information-theoretic lower bound for the episodic MDP setting studied in this paper:

**Theorem 4.** *For the episodic MDP problem studied in this paper, the expected regret for any algorithm must be at least $\Omega(\sqrt{H^2 SAT})$.*

Theorem 4 (see Appendix D for details) shows that both variants of our algorithm are nearly optimal, in the sense they differ from the optimal regret by a factor of $H$ and $\sqrt{H}$, respectively.

## 4 Proof for Q-learning with UCB-Hoeffding

In this section, we provide the full proof of Theorem 2. Intuitively, the episodic MDP with $H$ steps per epsiode can be viewed as a contextual bandit of $H$ "layers." The key challenge here is to control the way error and confidence propagate through different "layers" in an online fashion, where our specific choice of exploration bonus and learning rate make the regret as sharp as possible.

**Notation.** We denote by $\mathbb{I}[A]$ the indicator function for event $A$. We denote by $(x_h^k, a_h^k)$ the actual state-action pair observed and chosen at step $h$ of episode $k$. We also denote by $Q_h^k, V_h^k, N_h^k$ respectively the $Q_h, V_h, N_h$ functions at the *beginning* of episode $k$. Using this notation, the update equation at episode $k$ can be rewritten as follows, for every $h \in [H]$:

$$Q_h^{k+1}(x, a) = \begin{cases} (1 - \alpha_t)Q_h^k(x, a) + \alpha_t[r_h(x, a) + V_{h+1}^k(x_{h+1}^k) + b_t] & \text{if } (x, a) = (x_h^k, a_h^k) \\ Q_h^k(x, a) & \text{otherwise .} \end{cases}$$
(4.1)

Accordingly,

$$V_h^k(x) \leftarrow \min \left\{ H, \max_{a' \in \mathcal{A}} Q_h^k(x, a') \right\}, \quad \forall x \in \mathcal{S} .$$

Recall that we have $[\mathbb{P}_h V_{h+1}](x, a) := \mathbb{E}_{x' \sim \mathbb{P}_h(\cdot|x,a)} V_{h+1}(x')$. We also denote its empirical counterpart of episode $k$ as $[\hat{\mathbb{P}}_h^k V_{h+1}](x, a) := V_{h+1}(x_{h+1}^k)$, which is defined only for $(x, a) = (x_h^k, a_h^k)$.

Recall that we have chosen the learning rate as $\alpha_t := \frac{H+1}{H+t}$. For notational convenience, we also introduce the following related quantities:

$$\alpha_t^0 = \prod_{j=1}^t (1 - \alpha_j), \qquad \alpha_t^i = \alpha_i \prod_{j=i+1}^t (1 - \alpha_j) .$$
(4.2)

It is easy to verify that (1) $\sum_{i=1}^{t} \alpha_t^i = 1$ and $\alpha_t^0 = 0$ for $t \geq 1$; (2) $\sum_{i=1}^{t} \alpha_t^i = 0$ and $\alpha_t^0 = 1$ for $t = 0$.

**Favoring Later Updates.** At any $(x, a, h, k) \in \mathcal{S} \times \mathcal{A} \times [H] \times [K]$, let $t = N_h^k(x, a)$ and suppose $(x, a)$ was previously taken at step $h$ of episodes $k_1, \ldots, k_t < k$. By the update equation (4.1) and the definition of $\alpha_t^i$ in (4.2), we have:

$$Q_h^k(x, a) = \alpha_t^0 H + \sum_{i=1}^{t} \alpha_t^i \left[ r_h(x, a) + V_{h+1}^{k_i}(x_{h+1}^{k_i}) + b_i \right] . \tag{4.3}$$

According to (4.3), the $Q$ value at episode $k$ equals a weighted average of the $V$ values of the "next states" with weights $\alpha_t^1, \ldots, \alpha_t^t$. Our choice of the learning rate $\alpha_t = \frac{H+1}{H+t}$ ensures that, approximately speaking, the last $1/H$ fraction of the indices $i$ is given non-negligible weights, whereas the first $1 - 1/H$ fraction is forgotten. This ensures that the information accumulates smoothly across the $H$ layers of the MDP. If one were to use $\alpha_t = \frac{1}{t}$ instead, the weights $\alpha_t^1, \ldots, \alpha_t^t$ would all equal $1/t$, and using those $V$ values from earlier episodes would hurt the accuracy of the $Q$ function. In contrast, if one were to use $\alpha_t = 1/\sqrt{t}$ instead, the weights $\alpha_t^1, \ldots, \alpha_t^t$ would concentrate too much on the most recent episodes, which would incur high variance.

## 4.1 Proof Details

We first present an auxiliary lemma which exhibits some important properties that result from our choice of learning rate. The proof is based on simple manipulations on the definition of $\alpha_t$, and is provided in Appendix B.

**Lemma 4.1.** *The following properties hold for $\alpha_t^i$:*

(a) $\frac{1}{\sqrt{t}} \leq \sum_{i=1}^{t} \frac{\alpha_t^i}{\sqrt{i}} \leq \frac{2}{\sqrt{t}}$ *for every $t \geq 1$.*

(b) $\max_{i \in [t]} \alpha_t^i \leq \frac{2H}{t}$ *and $\sum_{i=1}^{t} (\alpha_t^i)^2 \leq \frac{2H}{t}$ for every $t \geq 1$.*

(c) $\sum_{t=i}^{\infty} \alpha_t^i = 1 + \frac{1}{H}$ *for every $i \geq 1$.*

We note that property $(c)$ is especially important—as we will show later, each step in one episode can blow up the regret by a multiplicative factor of $\sum_{t=i}^{\infty} \alpha_t^i$. With our choice of learning rate, we ensure that this blow-up is at most $(1 + 1/H)^H$, which is a constant factor.

We now proceed to the formal proof. We start with a lemma that gives a recursive formula for $Q - Q^\star$, as a weighted average of previous updates.

**Lemma 4.2** (recursion on $Q$). *For any $(x, a, h) \in \mathcal{S} \times \mathcal{A} \times [H]$ and episode $k \in [K]$, let $t = N_h^k(x, a)$ and suppose $(x, a)$ was previously taken at step $h$ of episodes $k_1, \ldots, k_t < k$. Then:*

$$(Q_h^k - Q_h^\star)(x, a) = \alpha_t^0 (H - Q_h^\star(x, a)) + \sum_{i=1}^{t} \alpha_t^i \left[ (V_{h+1}^{k_i} - V_{h+1}^\star)(x_{h+1}^{k_i}) + [(\hat{\mathbb{P}}_h^{k_i} - \mathbb{P}_h)V_{h+1}^\star](x, a) + b_i \right] .$$

*Proof of Lemma 4.2.* From the Bellman optimality equation, $Q_h^\star(x, a) = (r_h + \mathbb{P}_h V_{h+1}^\star)(x, a)$, our notation $[\hat{\mathbb{P}}_h^{k_i} V_{h+1}](x, a) := V_{h+1}(x_{h+1}^{k_i})$, and the fact that $\sum_{i=0}^{t} \alpha_t^i = 1$, we have

$$Q_h^\star(x, a) = \alpha_t^0 Q_h^\star(x, a) + \sum_{i=1}^{t} \alpha_t^i \left[ r_h(x, a) + (\mathbb{P}_h - \hat{\mathbb{P}}_h^{k_i})V_{h+1}^\star(x, a) + V_{h+1}^\star(x_{h+1}^{k_i}) \right] .$$

Subtracting the formula (4.3) from this equation, we obtain Lemma 4.2. $\qquad\square$

Next, using Lemma 4.2 and the Azuma-Hoeffding concentration bound, our next lemma shows that $Q^k$ is always an upper bound on $Q^\star$ at any episode $k$, and the difference between $Q^k$ and $Q^\star$ can be bounded by quantities from the next step.

**Lemma 4.3** (bound on $Q^k - Q^\star$). *There exists an absolute constant $c > 0$ such that, for any $p \in (0, 1)$, letting $b_t = c\sqrt{H^3 \iota / t}$, we have $\beta_t = 2 \sum_{i=1}^{t} \alpha_t^i b_i \leq 4c\sqrt{H^3 \iota / t}$ and, with probability*

*at least $1 - p$, the following holds simultaneously for all $(x, a, h, k) \in \mathcal{S} \times \mathcal{A} \times [H] \times [K]$:*

$$0 \leq (Q_h^k - Q_h^\star)(x, a) \leq \alpha_t^0 H + \sum_{i=1}^t \alpha_t^i (V_{h+1}^{k_i} - V_{h+1}^\star)(x_{h+1}^{k_i}) + \beta_t \ ,$$

*where $t = N_h^k(x, a)$ and $k_1, \ldots, k_t < k$ are the episodes where $(x, a)$ was taken at step $h$.*

*Proof of Lemma 4.3.* For each fixed $(x, a, h) \in \mathcal{S} \times \mathcal{A} \times [H]$, let us denote $k_0 = 0$, and denote

$$k_i = \min \left( \{ k \in [K] \mid k > k_{i-1} \wedge (x_h^k, a_h^k) = (x, a) \} \cup \{ K + 1 \} \right) \ .$$

That is, $k_i$ is the episode of which $(x, a)$ was taken at step $h$ for the $i$th time (or $k_i = K + 1$ if it is taken for fewer than $i$ times). The random variable $k_i$ is clearly a stopping time. Let $\mathcal{F}_i$ be the $\sigma$-field generated by all the random variables until episode $k_i$, step $h$. Then, $\left( \mathbb{I}[k_i \leq K] \cdot [(\hat{\mathbb{P}}_h^{k_i} - \mathbb{P}_h) V_{h+1}^\star](x, a) \right)_{i=1}^\tau$ is a martingale difference sequence w.r.t the filtration $\{ \mathcal{F}_i \}_{i \geq 0}$. By Azuma-Hoeffding and a union bound, we have that with probability at least $1 - p/(SAH)$:

$$\forall \tau \in [K]: \quad \left| \sum_{i=1}^\tau \alpha_\tau^i \cdot \mathbb{I}[k_i \leq K] \cdot [(\hat{\mathbb{P}}_h^{k_i} - \mathbb{P}_h) V_{h+1}^\star](x, a) \right| \leq \frac{cH}{2} \sqrt{\sum_{i=1}^\tau (\alpha_\tau^i)^2 \cdot \iota} \leq c \sqrt{\frac{H^3 \iota}{\tau}} \ , \tag{4.4}$$

for some absolute constant $c$. Because inequality (4.4) holds for all fixed $\tau \in [K]$ uniformly, it also holds for $\tau = t = N_h^k(x, a) \leq K$, which is a random variable, where $k \in [K]$. Also note $\mathbb{I}[k_i \leq K] = 1$ for all $i \leq N_h^k(x, a)$. Putting everything together, and using a union bound, we see that with least $1 - p$ probability, the following holds simultaneously for all $(x, a, h, k) \in \mathcal{S} \times \mathcal{A} \times [H] \times [K]$:

$$\left| \sum_{i=1}^t \alpha_t^i [(\hat{\mathbb{P}}_h^{k_i} - \mathbb{P}_h) V_{h+1}^\star](x, a) \right| \leq c \sqrt{\frac{H^3 \iota}{t}} \quad \text{where} \quad t = N_h^k(x, a) \ . \tag{4.5}$$

On the other hand, if we choose $b_t = c\sqrt{H^3 \iota / t}$ for the same constant $c$ in Eq. (4.4), we have $\beta_t / 2 = \sum_{i=1}^t \alpha_t^i b_i \in [c\sqrt{H^3 \iota / t}, 2c\sqrt{H^3 \iota / t}]$ according to Lemma 4.1.a. Then the right-hand side of Lemma 4.3 follows immediately from Lemma 4.2 and inequality (4.5). The left-hand side also follows from Lemma 4.2 and Eq. (4.5) and induction on $h = H, H - 1, \ldots, 1$. $\qquad\square$

We are now ready to prove Theorem 2. The proof decomposes the regret in a recursive form, and carefully controls the error propagation with repeated usage of Lemma 4.3.

*Proof of Theorem 2.* Denote by

$$\delta_h^k := (V_h^k - V_h^{\pi_k})(x_h^k) \quad \text{and} \quad \phi_h^k := (V_h^k - V_h^\star)(x_h^k) \ .$$

By Lemma 4.3, we have that with $1 - p$ probability, $Q_h^k \geq Q_h^\star$ and thus $V_h^k \geq V_h^\star$. Thus, the total regret can be upper bounded:

$$\text{Regret}(K) = \sum_{k=1}^K (V_1^\star - V_1^{\pi_k})(x_1^k) \leq \sum_{k=1}^K (V_1^k - V_1^{\pi_k})(x_1^k) = \sum_{k=1}^K \delta_1^k \ .$$

The main idea of the rest of the proof is to upper bound $\sum_{k=1}^K \delta_h^k$ by the next step $\sum_{k=1}^K \delta_{h+1}^k$, thus giving a recursive formula to calculate total regret. We can obtain such a recursive formula by relating $\sum_{k=1}^K \delta_h^k$ to $\sum_{k=1}^K \phi_h^k$.

For any fixed $(k, h) \in [K] \times [H]$, let $t = N_h^k(x_h^k, a_h^k)$, and suppose $(x_h^k, a_h^k)$ was previously taken at step $h$ of episodes $k_1, \ldots, k_t < k$. Then we have:

$$\begin{aligned}
\delta_h^k = (V_h^k - V_h^{\pi_k})(x_h^k) &\overset{①}{\leq} (Q_h^k - Q_h^{\pi_k})(x_h^k, a_h^k) \\
&= (Q_h^k - Q_h^\star)(x_h^k, a_h^k) + (Q_h^\star - Q_h^{\pi_k})(x_h^k, a_h^k) \\
&\overset{②}{\leq} \alpha_t^0 H + \sum_{i=1}^t \alpha_t^i \phi_{h+1}^{k_i} + \beta_t + [\mathbb{P}_h(V_{h+1}^\star - V_{h+1}^{\pi_k})](x_h^k, a_h^k) \\
&\overset{③}{=} \alpha_t^0 H + \sum_{i=1}^t \alpha_t^i \phi_{h+1}^{k_i} + \beta_t - \phi_{h+1}^k + \delta_{h+1}^k + \xi_{h+1}^k \ ,
\end{aligned} \tag{4.6}$$

where $\beta_t = 2 \sum \alpha_t^i b_i \leq O(1)\sqrt{H^3 \iota / t}$ and $\xi_{h+1}^k := [(\mathbb{P}_h - \hat{\mathbb{P}}_h^k)(V_{h+1}^\star - V_{h+1}^k)](x_h^k, a_h^k)$ is a martingale difference sequence. Inequality ① holds because $V_h^k(x_h^k) \leq \max_{a' \in \mathcal{A}} Q_h^k(x_h^k, a') =$

$Q_h^k(x_h^k, a_h^k)$, and inequality ② holds by Lemma 4.3 and the Bellman equation (2.1). Finally, equality ③ holds by definition $\delta_{h+1}^k - \phi_{h+1}^k = (V_{h+1}^\star - V_{h+1}^{\pi_k})(x_{h+1}^k)$.

We turn to computing the summation $\sum_{k=1}^K \delta_h^k$. Denoting by $n_h^k = N_h^k(x_h^k, a_h^k)$, we have:

$$\sum_{k=1}^K \alpha_{n_h^k}^0 H = \sum_{k=1}^K H \cdot \mathbb{I}[n_h^k = 0] \le SAH \ .$$

The key step is to upper bound the second term in (4.6), which is:

$$\sum_{k=1}^K \sum_{i=1}^{n_h^k} \alpha_{n_h^k}^i \phi_{h+1}^{k_i(x_h^k, a_h^k)},$$

where $k_i(x_h^k, a_h^k)$ is the episode in which $(x_h^k, a_h^k)$ was taken at step $h$ for the $i$th time. We regroup the summands in a different way. For every $k' \in [K]$, the term $\phi_{h+1}^{k'}$ appears in the summand with $k > k'$ if and only if $(x_h^k, s_h^k) = (x_h^{k'}, s_h^{k'})$. The first time it appears we have $n_h^k = n_h^{k'} + 1$, the second time it appears we have $n_h^k = n_h^{k'} + 2$, and so on. Therefore

$$\sum_{k=1}^K \sum_{i=1}^{n_h^k} \alpha_{n_h^k}^i \phi_{h+1}^{k_i(x_h^k, a_h^k)} \le \sum_{k'=1}^K \phi_{h+1}^{k'} \sum_{t=n_h^{k'}+1}^\infty \alpha_t^{n_h^{k'}} \le \left(1 + \frac{1}{H}\right) \sum_{k=1}^K \phi_{h+1}^k,$$

where the final inequality uses $\sum_{t=i}^\infty \alpha_t^i = 1 + \frac{1}{H}$ from Lemma 4.1.c. Plugging these back into (4.6), we have:

$$\sum_{k=1}^K \delta_h^k \le SAH + \left(1 + \frac{1}{H}\right) \sum_{k=1}^K \phi_{h+1}^k - \sum_{k=1}^K \phi_{h+1}^k + \sum_{k=1}^K \delta_{h+1}^k + \sum_{k=1}^K (\beta_{n_h^k} + \xi_{h+1}^k)$$

$$\le SAH + \left(1 + \frac{1}{H}\right) \sum_{k=1}^K \delta_{h+1}^k + \sum_{k=1}^K (\beta_{n_h^k} + \xi_{h+1}^k) \ , \tag{4.7}$$

where the final inequality uses $\phi_{h+1}^k \le \delta_{h+1}^k$ (owing to the fact that $V^\star \ge V^{\pi_k}$). Recursing the result for $h = 1, 2, \ldots, H$, and using the fact $\delta_{H+1}^K \equiv 0$, we have:

$$\sum_{k=1}^K \delta_1^k \le O\left(H^2 SA + \sum_{h=1}^H \sum_{k=1}^K (\beta_{n_h^k} + \xi_{h+1}^k)\right).$$

Finally, by the pigeonhole principle, for any $h \in [H]$:

$$\sum_{k=1}^K \beta_{n_h^k} \le O(1) \cdot \sum_{k=1}^K \sqrt{\frac{H^3 \iota}{n_h^k}} = O(1) \cdot \sum_{x,a} \sum_{n=1}^{N_h^K(x,a)} \sqrt{\frac{H^3 \iota}{n}} \overset{①}{\le} O\left(\sqrt{H^3 SAK\iota}\right) = O\left(\sqrt{H^2 SAT\iota}\right) \tag{4.8}$$

where inequality ① is true because $\sum_{x,a} N_h^K(x,a) = K$ and the left-hand side of ① is maximized when $N_h^K(x,a) = K/SA$ for all $x, a$. Also, by the Azuma-Hoeffding inequality, with probability $1 - p$, we have:

$$\left|\sum_{h=1}^H \sum_{k=1}^K \xi_{h+1}^k\right| = \left|\sum_{h=1}^H \sum_{k=1}^K [(\mathbb{P}_h - \hat{\mathbb{P}}_h^k)(V_{h+1}^\star - V_{h+1}^k)](x_h^k, a_h^k)\right| \le cH\sqrt{T\iota}.$$

This establishes $\sum_{k=1}^K \delta_1^k \le O\left(H^2 SA + \sqrt{H^4 SAT\iota}\right)$. We note that when $T \ge \sqrt{H^4 SAT\iota}$, we have $\sqrt{H^4 SAT\iota} \ge H^2 SA$, and when $T \le \sqrt{H^4 SAT\iota}$, we have $\sum_{k=1}^K \delta_1^k \le HK = T \le \sqrt{H^4 SAT\iota}$. Therefore, we can remove the $H^2 SA$ term in the regret upper bound.

In sum, we have $\sum_{k=1}^K \delta_1^k \le O\left(H^2 SA + \sqrt{H^4 SAT\iota}\right)$, with probability at least $1 - 2p$. Rescaling $p$ to $p/2$ finishes the proof. $\square$

## Acknowledgements

We thank Nan Jiang, Sham M. Kakade, Greg Yang and Chicheng Zhang for valuable discussions. This work was supported in part by the DARPA program on Lifelong Learning Machines, and Microsoft Research Gratis Traveler program.

## Footnotes

[1]Jaksch et al. [10] and Agrawal and Jia [1] apply to the more general setting of weakly communicating MDPs with $S'$ states and diameter $D$; our episodic MDP is a special case obtained by augmenting the state space so that $S' = SH$ and $D \geq H$.

[2]Azar et al. [5] and Kakade et al. [12] assume equal transition matrices $\mathbb{P}_1 = \cdots = \mathbb{P}_H$; in the setting of this paper $\mathbb{P}_1, \cdots, \mathbb{P}_H$ can be entirely different. This adds a factor of $\sqrt{H}$ to their total regret.

[3]Strehl et al. [25] applies to MDPs with $S'$ states and discount factor $\gamma$; our episodic MDP can be converted to that case by setting $S' = SH$ and $1 - \gamma = 1/H$. Their result only applies to the stochastic setting where initial states $x_1^k$ come from a fixed distribution, and only gives a PAC guarantee. See our full version for a comparison between PAC and regret guarantees.

[4]While we study deterministic reward functions for notational simplicity, our results generalize to randomized reward functions. Also, we assume the reward is in $[0, 1]$ without loss of generality.

[5] Recall that for independent zero-mean random variables $X_1, \ldots, X_T$ satisfying $|X_i| \leq M$, their summation does not exceed $\tilde{\mathcal{O}}(M\sqrt{T})$ with high probability using Hoeffding concentration. If we have in hand a better variance bound, this can be improved to $\tilde{\mathcal{O}}\left(M + \sqrt{\sum_i \mathbb{E}[X_i]^2}\right)$ using Bernstein concentration.

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
