[Reviews · NeurIPS 2018]

Reviewer 1



Is Q-learning Provably Efficient? ================================= This paper studies the problem of efficient exploration in finite episodic MDPs. They present a variant of optimistic initialization + tuned learning rates for Q-learning that recover a UCB-style algorithm. The main contribution of this work is a polynomial regret bound for perhaps one of the most iconic "model-free" algorithms. There are several things to like about this paper: - Q-learning is perhaps the classic intro to RL algorithms, so it's nice to see that we can recover sample efficient guarantees for a variant of this algorithm. - The writing/structure of the paper is clear and appealing; particularly given a potentially dry/analytical subject matter - good job! - There is some genuine new result in this paper, the first "worst-case" sqrt{T} regret bounds for a model-free RL algorithm + a clear exposition. The computational time is also particularly appealing compared to existing model-free algorithms with sqrt{T} *expected* (Bayesian) regret (such as RLSVI), which have much higher computational and memory requirements. - Trading off the statistical complexity (regret) with the computational complexity (time/space) is a clearly valuable element of the discussion in this literature, and one that is often overlooked. The concise presentation in Table 1 is laudable and will help to clarify the field for many. However, the paper also has a few shortcomings: - The bounds are *not* the first sqrt{T} bounds for model-free reinforcement learning... at least not according to the definition offered in the first paragraph of this paper. That distinction goes to "randomized least squares value iteration" from "Generalization and Exploration via Randomized Value Functions" see https://arxiv.org/abs/1402.0635 or more complete presentation/analysis in https://arxiv.org/abs/1703.07608. It should be noted that that analysis is given in terms of the Bayesian Expected regret, not the worst case regret as studied here... but if you want to claim the first sqrt{T} bounds then I think it is very important to qualify this technical distinction. It is interesting to provide a similar algorithm with worst-case guarantees... but I don't think it's as big a jump. Note too that the quality of these bounds esentially matches that of RLSVI.... but unlike RLSVI this algorithm does not naturally extend to generalization. - The point above greatly affects the novelty and impact of this paper. In addition, most of the technical results / analysis approach is really derived from a similar line of work, that begins with RLSVI -> UCBVI -> This Q-learning analysis. - Note, on the positive side here, that there are potential other definitions of memory/runtime that do distinguish this analysis and mean that, within *that* class of models this is the first provably-efficient application of UCB and this is a great contribution especially given the central status of Q-learning! - The distinction between model-free and model-based in this tabular context is not really as big a distinction as the paper makes out. In many ways this paper feels like a way of relabelling the memory requirements of UCBVI [5] by separating each estimation problem to be independent at each timestep. Most of the tools seem to be repurposed from the "standard library". The real benefit seems to come from computational savings that are afforded by the extra H-dependence. - Actually this issue of learning a distinct Q for each "timestep" and particular NOT SHARING DATA across timesteps sidesteps one of the key difficulties in the analysis of UCBVI + related algorithms. That's OK, but it feels like the paper is hiding some of that, whereas I think it should take center stage to help explain what is going on here. - Q-learning is interesting to study particularly because of its connection to DQN... but I'm not convinced that this paper really gives us much insight that we can carry over to RL with function approximation. I think "practioners" already knew that UCB is theoretically better than e-greedy and I'm really not convinced by the suggestion that practitioners use a learning rate O(H) at the start... I think that only applies when you have a tabular Q separated at each timestep... this "key takeaway" could even be damaging if it is wrong! The interesting setting is Q learning with function approximation, but this presentation is misleading if it makes people think Q-learning with function approximation is efficient. Furthermore, it is well known that *everyone* using neural networks typically tunes the learning rate over orders of magnitude, so I'm not sure that this claim of O(H/t) vs O(1/t) really affords anything interesting. - The other observation that UCB affords better treatment of uncertainty is better than epsilon-greedy is also essentially vacuous... epsilon-greedy doesn't maintain any treatment of uncertainty! Overall I think that this is a reasonable and even good paper in many parts. I've gone back to this review more than any other paper in my batch of reviews and in many ways it does leave me torn: There are significant issues in the novelty, particularly in their claim to be the first sqrt{T} regret bound model-free RL algorithm - which is incorrect according to the paper's definitions and thus cannot be accepted in its current state. Many of the other claims are overblown or even misleading to "practitioners", where generalization should of course be a primary concern. On the other hand, if the authors can address some of the objections then I think it could be further improved and it will be a great candidate for another conference or potentially here. The existence of previous "model-free" algorithms does not negate the contribution of this work and, with respect to time and space, this is a clear improvement for the tabular finite horizon setting. Due to the issues I have raised I have sided with a rejection... I think there are just too many issues and too essential a rewrite to recommend acceptance without access to the camera-ready version. However, I feel confident that it *will* be possible to address these issues for a future version and that this will help to make it a better paper. I've indicated a 5/10 as hope to reflect this dual-sided view. ========================================== Amended post-rebuttal Thank you to the authors for your thoughtful rebuttal. I really agree with the points that you make... and I just want to point out that this type of discussion is *not* present in the version of the paper you submitted. My worries are not based in whether or not this paper can/should be a great contribution - I think so, more that the way it is currently presented is potentially misleading. For example, the proposed definition of model-free differs greatly from what is currently in the paper and that's important. I think that a discussion on this topic is particularly valuable and helps to clarify the contribution of this work. - For RLSVI I think it would be helpful to mention the complexity in terms of D=dimension of approximating basis functions, but of course the only case with proven regret bounds are for D=SA. On the topic of distinct Q and RL with function approximation, it's not so much that I think it's unworthy to study this simpler situation - I think it's a great first step. More, that I worry the claims for "practitioner" who applies Q-learning with neural network and discounted problem might be overstated. I'm not sure that the same conclusions apply in this setting, what does it mean to do optimistic initialization for neural networks + SGD? My worry following the rebuttal is how many of these points / amendments would actually make it into a final camera ready version. However, under the understanding that the area chair would enforce this requirement I will upgrade my review to a 6.

Reviewer 2



SUMMARY This paper considers an optimistic variant of Q-learning in an episodic reinforcement learning setting where the transition matrix at each stage is fixed but may be different from the transition matrix of the other stages. Regret bounds are derived that match a lower bound derived in this particular setting up to a factor of sqrt{H}, where H is the episode horizon. EVALUATION This is an ok paper that adapts the episodic RL analysis of Azar et al (ICML 2017) to an optimistic Q-learning variant and generalizes it to the case where transition probabilities may be different at different stages. Due to the latter particularity the lower bound that can be achieved is worse than the lower bound in Azar et al (ICML 2017) - which should be pointed out in the main part of the paper and not only in the appendix however. As far as I can tell, apart from the (rather artifical) generalization about the stage-dependent transitions, the analysis is quite similar to Azar et al (ICML 2017). Due to the more general setting this gives regret bounds that are worse by a factor of sqrt{H}. In my opinion the authors try to oversell their results however. First of all, while the title suggests an analysis of Q-learning, the paper considers a particular variant of Q-learning with exploration bonus and a predefined fixed learning rate, and the analysis is restricted to a rather particular episodic setting. The paper claims to give the first analysis of a model-free RL algorithm, yet does not provide a clear definition what precisely distinguishes a model-free from a model-based algorithm. The vague claim that "model-free algorithms (...) directly update the value function" may as well apply to the original paper of Azar et al (ICML 2017), which employs optimistic Q- and value-function, too. Finally, the comparison to other algorithms is rather unfair. In particular, mixing up results for the episodic and the non-episodic case the way it is done in Table 1 is in my view highly misleading. Moreover, it is obvious that if the other algorithms used for comparison would be given the same information as the Q-learning algorithm considered in the paper uses (i.e., that transitions are the same at the same stage) would perform better than claimed in Table 1. Overall, in my view the paper's main shortcoming is that it does not discuss the differences between the given optimistic Q-learning algorithm and the algorithm of Azar et al (ICML 2017). I think it would be necessary to argue that the latter algorithm cannot be rewritten as a Q-learning variant. Otherwise, the paper at hand would add little to the contribution of Azar et al (ICML 2017). COMMENTS - I did not see why when considering the episodic setting as a special case of MDPs (footnote 1 on p.3) the diameter D=H. Depending on how easy it is to reach a particular state, I think the diameter may be much larger than that. - In the algorithm, p should be given as parameter. The alpha's as given in (3.1) should be specified either in the algorithm or in Thm 1. - In l.149, there seems to be missing something in the last part of the sentence. - Thm 3: As said, it should be explicitly mentioned here why this result is not in contradiction to the paper of Azar et al (ICML 2017). - While I believe that one can derive PAC bounds from the regret bounds, I could not follow the argument between lines 171 and 172. In particular, I could not see why random policy selection would be considered for that. - I found notation a bit unusual. Typically, t denotes a time step, and delta would be a more common notation for the error probability. - The paper uses two different *-notations. - There are a few typos, missing words (mainly articles and 'that'). In general. the paper should be proof-read for English style and grammar issues. --- POST-REBUTTAL I acknowledge the definition of model-free and model-based algorithms given in the author feedback and that the presented algorithm has an improved space complexity over UCBVI. Still, in my opinion this is not such a significant advantage, in particular (as Reviewer #1 mentions) in the tabular setting. Also, it is still not clear to me that a reformulation of UCBVI could not give model-free space complexity, too. Given that the paper considers a (for the addressed practitioners) hardly relevant episodic setting, my overall assessment does not change for the moment.

Reviewer 3



This paper proves near-optimal regret bounds for Q-learning with UCB-bonus style exploration in the tabular, episodic MDP setting. I really like how the algorithm is just Q-learning with a simple annealing learning rate and simple exploration bonus. It's much simpler than Delayed Q-learning. ==== After Author Rebuttal ==== As I'm not as intimately familiar with the proof of UCBVI, I cannot determine how straightforward it would be to come up with a version of UCBVI that has reduced space complexity. I agree with the other authors that some claims are perhaps overblown; however I do think that the reduced space complexity is novel (unless I'm missing some other prior work). Even though this algorithm is not just Q-learning since it is using UCB exploration, I do think that showing the Q-learning-style updates can still result in efficient learning is interesting. I would tend to keep my decision of acceptance, but this could change as I am not confident in my judge of novelty.